# Using Urine Biomarkers to Differentiate Bladder Dysfunctions in Women with Sensory Bladder Disorders

**DOI:** 10.3390/ijms25179359

**Published:** 2024-08-29

**Authors:** Yu-Chen Chen, Yuan-Hong Jiang, Jia-Fong Jhang, Hann-Chorng Kuo

**Affiliations:** 1Graduate Institute of Clinical Medicine, College of Medicine, Kaohsiung Medical University, Kaohsiung 80756, Taiwan; jennis7995@hotmail.com; 2Department of Urology, Kaohsiung Medical University Hospital, Kaohsiung Medical University, Kaohsiung 80756, Taiwan; 3Regenerative Medicine and Cell Therapy Research Center, Kaohsiung Medical University, Kaohsiung 80756, Taiwan; 4Department of Urology, Hualien Tzu Chi Hospital, Buddhist Tzu Chi Medical Foundation, Tzu Chi University, Hualien 97004, Taiwan; redeemerhd@gmail.com (Y.-H.J.); alur1984@hotmail.com (J.-F.J.)

**Keywords:** cystitis, urine biomarker, bladder inflammation, overactive bladder, bladder pain syndrome

## Abstract

Sensory bladder disorders encompass several distinct conditions with overlapping symptoms, which pose diagnostic challenges. This study aimed to evaluate urine biomarkers for differentiating between various sensory bladder disorders, including non-Hunner’s interstitial cystitis (NHIC), detrusor overactivity (DO), hypersensitive bladder (HSB), and urodynamically normal women. A retrospective analysis of 191 women who underwent a videourodynamic study (VUDS) was conducted, with some also receiving cystoscopic hydrodistention to confirm the presence of NHIC. Participants were categorized into four groups: DO (n = 51), HSB (n = 29), NHIC (n = 81), and normal controls (n = 30). The urine levels of inflammatory and oxidative stress biomarkers were measured. The DO patients exhibited elevated IP-10 levels, while the HSB patients had decreased TAC and 8-OHdG levels. The NHIC patients showed lower IL-2 and higher TNF-α levels. A TNF-α ≥ 1.05 effectively identified NHIC, with an AUROC of 0.889, a sensitivity of 98.8%, and a specificity of 81.3%. An IP-10 ≥ 6.31 differentiated DO with an AUROC of 0.695, a sensitivity of 56.8%, and a specificity of 72.3%. An 8-OHdG ≤ 14.705 and a TAC ≤ 528.7 identified HSB with AUROCs of 0.754 and 0.844, respectively. The combination of 8-OHdG and TAC provided an AUROC of 0.853 for HSB. These findings suggest that TNF-α, IP-10, TAC, 8-OHdG, and IL-2 are promising non-invasive biomarkers for distinguishing between these conditions, which may improve diagnosis and management.

## 1. Introduction

Sensory bladder disorders are common lower urinary tract dysfunctions related to abnormal bladder sensations encountered in urological clinics, encompassing conditions such as detrusor overactivity (DO), interstitial cystitis/bladder pain syndrome (IC/BPS), and hypersensitive bladder (HSB) [1]. These disorders also include women with normal bladder function who have an overactive bladder (OAB) due to anxiety. Patients with sensory bladder disorders typically present with symptoms such as frequency, nocturia, urgency, lower abdominal discomfort, occasional urgency urinary incontinence, or voiding difficulty. Due to the overlapping symptoms in these sensory bladder disorders, differential diagnosis among them is challenging.

Current diagnostic evaluations and procedures, such as uroflowmetry, post-void residual measurement, and cystoscopy, are primarily focused on structural and functional attributes [2]. However, these methods do not adequately address the underlying biochemical and pathophysiological mechanisms unique to each disorder, often leading to the need for more invasive, costly, and time-consuming procedures like urodynamic studies and cystoscopic hydrodistention for accurate diagnosis. Standard treatments involve the use of antimuscarinics or beta-3 adrenoceptor agonists, which often yield only partial relief [2,3]. Urodynamic studies may be recommended when initial medical treatments fail [4], and cystoscopic hydrodistention might be suggested to confirm IC/BPS presence [5]. Therefore, developing non-invasive, biochemical-based diagnostic options is crucial for improving management and outcomes for these patients.

The sensory nerve activation in sensory bladder disorders suggests that bladder sensory symptoms indicate specific pathways provoking sensory receptors and nerves [6,7]. Because the pathophysiology of sensory bladder disorders could be neurogenic, inflammatory, bladder outlet obstructive, vasculogenic, or oxidative stress, some urinary proteins and biomarkers might be elevated in specific lower urinary tract dysfunctions, providing a potential diagnostic avenue [7,8,9,10]. By identifying elevated levels of specific urinary biomarkers, it is feasible to apply a simple urine test as an initial diagnostic tool. Following the exclusion of urinary tract infections (UTIs), this approach could lead to earlier intervention and personalized management of sensory bladder disorders. Recent studies, such as one that investigates the relationship between urinary microbiomes and bladder cancer, suggest the broader applicability of these biomarkers in detecting other urological conditions [11]. This highlights the diagnostic potential of urinary biomarkers not only for sensory bladder disorders but also for predicting oncological risks, suggesting that these markers could form part of a larger panel for a comprehensive urological assessment.

This study aims to evaluate the potential urinary biomarkers for distinguishing NHIC, DO, and HSB among women with sensory bladder disorders. By comparing urinary inflammatory, neurogenic, and oxidative stress biomarkers in patients with videourodynamic-confirmed DO, HSB, normal controls with no specific findings, and NHIC confirmed via cystoscopic hydrodistention under anesthesia, we aim to identify specific urinary biomarkers that can effectively discriminate between these sensory bladder disorders. This approach is particularly relevant for patients with HSB and NHIC, thereby improving diagnostic accuracy and treatment strategies.

## 2. Results

A total of 191 women with newly diagnosed bladder sensory disorders were included. They were categorized into four groups based on VUDS results: NHIC (n = 81), DO (n = 51), HSB (n = 29), and controls (n = 30). The mean age was 58.7 ± 12.6 years, although patients with NHIC were significantly younger than patients with DO and HSB (Table 1).

### 2.1. Different Levels of Urinary Biomarkers across Groups

Significant differences were observed in urine biomarkers among the different groups (Table 1). Patients with NHIC had significantly lower levels of IL-2 and higher levels of TNF-α compared to the other groups. Patients with DO exhibited significantly higher levels of IP-10 and MIP-1β. Patients with HSB had significantly lower levels of TAC and 8-OHdG compared to the NHIC and control groups. The ratio of each significant urine biomarker level in sensory bladder disorder groups to control is shown in Figure 1.

### 2.2. Different VUDS Parameters across Groups

Significant differences in the VUDS parameters were also noted among the groups (Table 2). The DO group showed a significantly higher Pdet and lower compliance compared to the other groups. The control group had significantly higher Qmax and CBC compared to the DO and NHIC groups. The HSB patients demonstrated higher FSF, FS, and CBC compared to the DO and NHIC groups. The NHIC group had a significantly higher PVR and a lower bladder contractility index compared to controls.

### 2.3. The Diagnostic Value of Urine Biomarkers

The receiver operating characteristic (ROC) curve analysis identified key biomarkers with high diagnostic value (Figure 2). The ROC curve analysis identified TNF-α ≥ 1.05 as the optimal cut-off point for distinguishing NHIC from total patients, with an AUROC of 0.889, a sensitivity of 98.8%, and a specificity of 81.3%. For identifying patients with DO, a cut-off value of IP-10 ≥ 6.310 had an area under the ROC curve (AUROC) of 0.695, a sensitivity of 57.1%, and a specificity of 72.3%. An 8-OHdG ≤ 14.705 had an AUROC of 0.754 in identifying HSB, with a sensitivity of 87.5% and a specificity of 62.0%, whereas a TAC ≤ 528.7 had an AUROC of 0.844, a sensitivity of 100%, and a specificity of 59.5% for HSB. The combination of 8-OHdG and TAC had the highest predictive value for identifying HSB from total patients, with an AUROC of 0.853, a sensitivity of 100%, and a specificity of 68.6%.

### 2.4. Correlation Analysis between Biomarkers and VUDS Parameters

We compared the associations between different urinary biomarkers and VUDS parameters across the four groups. Significant strong associations (defined as a coefficient > 0.6) were observed between the urinary biomarkers and the VUDS parameters in the control group. Notably, PVR showed a positive correlation with TNF-α (γ = 0.874) and IL-2 (γ = 0.614). Voiding efficacy demonstrated a negative correlation with TNF-α (γ = −0.866). Some moderate associations (defined as a coefficient between 0.4 and 0.6) were also found across the control, DO, and HSB groups (Figure 3).

## 3. Discussion

This study conducted a comprehensive assessment of urinary biomarkers and VUDS parameters in the diagnosis of different bladder sensory disorders, namely NHIC, DO, and HSB, against urodynamically normal controls, yielding novel findings. Specifically, we observed the following: (1) significant disparities in urinary biomarkers across groups, including IL-2, TNF-α, IP-10, TAC, 8-OHdG, and MIP-1β. (2) TNF-α can serve as an effective and reliable diagnostic marker for differentiating NHIC from the other bladder sensory disorders. When a TNF-α level is ≥1.05, we may recommend bladder sensory disorder patients undergo cystoscopic hydrodistention under anesthesia for further diagnosis of NHIC (AUROC of 0.889, sensitivity of 98.8%, and specificity of 81.3%). (3) Using 8-OHdG and TAC together is an effective diagnostic tool for identifying patients with HSB among patients with bladder sensory disorders. When 8-OHdG is ≤14.705 and TAC is ≤528.72, the diagnostic sensitivity for HSB is 87.5% and the specificity is 79.5%. This combination of biomarkers provides a robust method for distinguishing HSB from the other bladder sensory disorders.

Bladder sensory disorders in women often present as a constellation of storage and/or voiding LUTS and/or bladder discomfort sensation, which may indicate various diseases, such as OAB or DO, diagnosed by urodynamic studies, or IC/BPS [12]. Therefore, further evaluation is often required to accurately diagnose the bladder condition and tailor appropriate treatment. These evaluations typically include bladder diaries, symptom-related questionnaires, PVR measurements, uroflowmetry, and sometimes urodynamic studies in complicated cases, or cystoscopic hydrodistention under anesthesia when IC/BPS is highly suspected [2]. However, these tests are time-consuming, require patient compliance and some are even invasive. Our goal was to determine whether non-invasive urinary biomarkers could assist in the differential diagnosis of bladder sensory disorders in female patients, thereby guiding subsequent treatment strategies more effectively.

Bladder sensory disorders are believed to arise from various pathophysiological mechanisms, including local bladder chronic inflammation, bladder urothelial dysfunction, and involvement of the nervous system leading to the hypersensitization of pain receptors or neurogenic inflammation-induced bladder discomfort [12,13,14,15]. Consequently, the expression levels of urinary inflammatory cytokines, chemokines, or oxidative stress markers could potentially elucidate disease mechanisms and serve as potential diagnostic indicators for LUTS [8,9,16,17,18,19,20,21,22]. Although several urinary biomarkers have been proposed, their clinical utility is limited due to their low sensitivity and specificity [13,23]. The 2024 EAU guidelines recommend against the routine use of urinary biomarkers in the diagnosis and management of lower urinary tract diseases, due to insufficient evidence regarding their diagnostic accuracy and validity in women [2]. This underscores the need for further research and the development of more accurate predictive biomarkers.

Our current study addresses this need by seeking more accurate predictive biomarkers that can be applied in clinical settings to distinguish between bladder sensory disorders presenting with similar symptoms. Our unique approach involved a cohort of 191 female patients with different bladder sensory disorders, excluding those with HIC and moderate and severe IC/BPS (glomerulation grade ≥ 2 and MBC < 760 mL under anesthesia), to create a more homogeneous study population who presented similar symptoms with the other sensory bladder disorders. By focusing on this specific group, we aimed to identify urinary biomarkers that could differentiate between these sensory bladder disorders and provide greater clinical utility. The findings of this study emphasize the diagnostic potential of specific urinary biomarkers, such as TNF-α, IL-2, 8-OHdG, and TAC, in managing sensory bladder disorders. Integrating these biomarkers into current clinical workflows could significantly enhance the accuracy of diagnosis and the effectiveness of treatment strategies in the future.

Our findings reveal distinct inflammatory profiles and oxidative stress biomarkers among women with different sensory bladder disorders. Specifically, the NHIC patients had significantly lower levels of IL-2 and higher levels of TNF-α compared to the other patient groups, indicating a unique inflammatory profile in the NHIC bladders. The patients with DO exhibited elevated levels of IP-10 and MIP-1β, while the HSB patients had lower levels of TAC and 8-OHdG compared to the NHIC and control groups. These differences in the expression levels suggest that specific urinary biomarkers could potentially aid in distinguishing between these sensory bladder disorders and provide insights into their underlying pathophysiological mechanisms. Moreover, compared to the existing literature, our findings corroborate the increasing recognition of TNF-α as a key biomarker for assessing bladder inflammation and sensory disorders, similar to the results reported by previous studies indicating high sensitivity and specificity for TNF-α in diagnosing various inflammatory conditions [22]. This connection reinforces the potential utility of TNF-α in clinical practice as a diagnostic tool.

In addition to exploring diagnostic biomarkers for sensory bladder disorders, it is pertinent to consider novel therapeutic interventions that may interact with or complement these biomarkers in clinical practice. For example, the use of intravesical platelet-rich plasma (PRP) injections has shown potential in treating IC/BPS [24]. This treatment demonstrates promise in enhancing urothelial regeneration and reducing chronic inflammation, which are key factors that may also be influenced by the urinary biomarkers we have identified. Such therapeutic advancements underscore the need for integrated diagnostic and treatment strategies, enriching the clinical relevance of our findings in urinary biomarkers.

This study highlights the potential diagnostic value of specific urinary biomarkers for distinguishing between different sensory bladder disorders. Our findings underscore that TNF-α shows high diagnostic accuracy, making it a reliable biomarker for identifying NHIC among women with sensory bladder disorders. This aligns with previous studies that have identified elevated oxidative stress biomarkers, such as 8-OHdG and 8-isoprostane, in IC/BPS patients compared to healthy controls, across all genders [9]. Additionally, research has shown significantly higher levels of inflammatory and pro-inflammatory cytokines, including eotaxin, MCP-1, and TNF-α, in male IC patients compared to other male LUTS conditions [19]. In women, MCP-1, RANTES, IP-10, IL-7, and eotaxin-1 have been found at elevated levels in ESSIC type 1 and type 2 IC/BPS compared to healthy controls [17]. Another study similarly noted that TNF-α and MIP-1β had an AUROC greater than 0.70 in predicting IC/BPS, including both Hunner-type IC/BPS and NHIC, compared with women with stress urinary incontinence [22]. Our study builds on these findings by providing new insights into the clinical application of TNF-α in women with sensory bladder disorders. Unlike previous studies [17,19,22], which primarily focused on broader or mixed populations, our research demonstrates that TNF-α can serve as a high-sensitivity and high-specificity tool for distinguishing NHIC from other bladder sensory disorders. Although AUA guidelines emphasize the careful exclusion of other LUTS possibilities when diagnosing IC/BPS [25], this process often requires time and additional tests, potentially delaying appropriate treatment for patients. A higher urine level of TNF-α offers an opportunity to streamline this diagnostic process. This provides a more confident diagnostic approach for clinicians, allowing them to promptly arrange cystoscopic hydrodistention under anesthesia for a definitive diagnosis in women with at least 6 weeks of bladder sensory symptoms and negative urine cultures.

Patients with DO usually present with cardinal symptoms of urgency and/or urgency urinary incontinence. However, for patients with DO without urinary incontinence, the perception of urgency might not be strong enough to distinguish it from other sensory bladder disorders, such as IC/BPS, OAB without urodynamic DO, or pure HSB. In this study, the urinary biomarker IP-10 was significantly higher in patients with DO compared to the other patient groups; however, the sensitivity was not high and the AUROC was only 0.695, indicating this urine biomarker might not be effective in differentiating DO from other sensory bladder disorders. Nevertheless, a higher urine level of IP-10 can still alert the physician to carefully monitor the patients’ LUTS and other objective urological findings, and to provide effective medication for DO.

We also identified the potential role of the combination of low urine levels of 8-OHdG and TAC in diagnosing HSB among patients with sensory bladder disorders. Our findings show that HSB patients had significantly lower levels of TAC and 8-OHdG compared to patients with NHIC or DO and controls. This is consistent with our previous studies, which have also reported lower TAC levels in HSB, indicating low oxidative stress [26]. Oxidative stress plays a significant role in LUTS-related diseases. It involves the production of reactive oxygen species due to tissue hypoxia, which can lead to damage to DNA, lipids, and proteins [26,27,28,29]. This damage can cause pathological changes in the bladder, resulting in sensory symptoms in patients with lower urinary tract dysfunctions. In this study, we have already excluded UTI, BOO, and other neurological diseases, therefore, the low oxidative stress urine biomarkers in patients with sensory bladder disorders could lead the physician to treat patients under the tentative diagnosis of bladder hypersensitivity but not the diagnosis of DO or IC/BPS.

These findings on the role of urinary biomarkers are supported by the significant variations observed in VUDS parameters across the different groups of sensory bladder disorders, reflecting the different pathophysiological mechanisms underlying these conditions. The strong correlations between the urinary biomarkers and the VUDS parameters, especially in the urodynamically normal group with bladder sensory symptoms, further emphasize the potential of these biomarkers in reflecting bladder function. The positive correlation between PVR and TNF-α and IL-2, along with the negative correlation between voiding efficacy and TNF-α, also indicate that inflammatory markers significantly influence bladder function. Previous studies have demonstrated that elevated urinary cytokines are present not only in storage LUTS-related diseases but also in voiding LUTS conditions [8,9,16,17,18]. This suggests that an inflamed bladder urothelial layer can contribute to both storage LUTS and bladder pain, as well as voiding LUTS [6,30]. Our data show that, in the urodynamically normal group with symptoms, the correlation between cytokine levels and bladder function is particularly strong. Although correlations are also present in the DO and NHIC groups, they are less pronounced. This could be due to the fact that storage LUTS, bladder pain, and voiding LUTS are all manifestations of bladder inflammation. Consequently, in the DO and NHIC groups, where cytokine levels are already elevated due to storage LUTS and bladder pain, it is challenging to isolate a strong correlation between individual urinary cytokines and voiding LUTS. While the exact extent to which inflammation contributes to storage versus voiding LUTS remains unclear, our findings provide a valuable foundation for future research. Understanding the interplay between different cytokines and bladder function can pave the way for more targeted therapeutic approaches and improve diagnostic accuracy for sensory bladder disorders.

This study has several limitations. First, no comorbidity control was made in our cohort. Different comorbidities could potentially affect urinary biomarker levels, although we have excluded neurogenic LUTS to minimize this impact. Second, our cohort only included women, so these findings might not apply to men. The causes of LUTS differ between males and females, so this study focuses solely on females. Third, our urinary biomarkers were not normalized with urinary creatinine. The debate surrounding the normalization of urine biomarkers with creatinine remains ongoing. We collected urine samples at a full bladder to minimize bias in urine biomarker density. Lastly, although our current study involved a single-center cohort, we acknowledge the importance of validating these biomarkers across more diverse populations. Further studies with well-controlled patient cohorts are necessary to validate these findings and explore the applicability of these biomarkers in broader populations.

## 4. Materials and Methods

This retrospective study included consecutive female patients who visited urological outpatient clinics with chief complaints of increased bladder sensation, urgency, or bladder discomfort, such as storage and/or voiding lower urinary tract symptoms (LUTS), lower abdominal discomfort, or bladder discomfort, from August 2012 to June 2021. This study was approved by the Institutional Review Board of Hualien Tzu Chi Hospital (IRB 111-102-B).

Inclusion criteria required patients to present with clinical symptoms of bladder sensory disorders. Exclusion criteria included the presence of an active UTI, urodynamic proven bladder outlet obstruction (BOO), intrinsic sphincter deficiency (ISD), or obvious neurological disorders. These criteria ensured a focused evaluation of sensory bladder dysfunction without confounding conditions such as infections or neurological influences. All patients were requested to keep a three-day voiding diary and record the daytime and nighttime voiding frequency, voided volume, episodes of urgency, and urgency incontinence (UUI). The maximal voided volume was recorded as the functional bladder capacity (FBC). When patients had an average daily frequency of more than eight times and a FBC less than 350 mL, bladder hypersensitivity was considered. When there were episodes of urgency or UUI, an overactive bladder was diagnosed.

### 4.1. Videourodynamic Study in the Diagnosis of Sensory Bladder Disorders

All included patients with bladder sensory disorders underwent a videourodynamic study (VUDS). The VUDS examination was performed with the patients in a sitting position. The infusion rate was set at 20 mL/min, and patients were carefully monitored for the bladder sensation during the filling phase, including the first sensation of filling (FSF), fullness sensation (FS), urgency sensation (US), and cystometric bladder capacity (CBC). The voiding detrusor pressure (Pdet), maximum flow rate (Qmax), voided volume, and post-void residual (PVR) volume parameters were also monitored. If involuntary detrusor contractions occurred during the filling phase or close to bladder capacity, phasic DO or terminal DO was diagnosed, respectively. Women with a Pdet of higher than 35 cmH_2_O and a narrow bladder outlet during the voiding phase were considered to have BOO [31]. Patients with urine leakage during the cough stress tests or Valsalva maneuver were considered to have ISD. After the VUDS, patients with a CBC of less than 350 mL were considered to have an increased bladder sensation, and a potassium chloride test was performed to elicit bladder pain and urgency. Patients with an increased bladder sensation but a negative KCl test were considered to have HSB. Patients with an increased bladder sensation and a positive KCl test were recommended to receive cystoscopic hydrodistention for the diagnosis of NHIC. Based on the VUDS findings, patients not diagnosed with NHIC were classified into DO and HSB. Patients who initially presented with sensory bladder disorder but had a CBC of more than 350 mL, without BOO or ISD, and normal voiding function were considered to have normal urodynamic findings and served as the control group. The technique of VUDS, terminology, and diagnoses were made according to the recommendations of the International Continence Society standards [32].

### 4.2. Cystoscopic Diagnosis of NHIC

For patients highly suspected of having Hunner’s IC (presenting with severe bladder pain and IC symptoms), office cystoscopy without anesthesia can usually identify Hunner’s lesion, and specific treatment is given [25,33,34]. For patients strongly suspected of having NHIC, cystoscopic hydrodistention under anesthesia and an intravesical pressure of 80 cmH_2_O were performed to check the maximal bladder capacity (MBC) and the grade of glomerulations developed after the release of bladder fluid, and to confirm the diagnosis of NHIC [25,34]. Patients were not included in this study if they had cystoscopic-confirmed Hunner’s IC, with a glomerulation grade of 2 or 3, or an MBC less than 760 mL, because their bladder inflammation was considered to be of moderate to severe degree [35]. All patients with NHIC included in this study had an MBC > 760 mL under anesthesia and a grade 0 or 1 glomerulation, indicating that they had very mild bladder inflammation and sensory bladder disorder. 

The primary aim of this study was to investigate the differences in urinary biomarkers and urodynamic parameters among women with different bladder sensory disorders, including NHIC, DO, HSB, and controls. The secondary aim was to evaluate the diagnostic value of specific urinary biomarkers for distinguishing between these bladder sensory disorder subtypes and to explore the correlations between these urinary biomarkers and VUDS parameters.

### 4.3. Urine Biomarkers Measurement

Urine samples were collected from all patients prior to VUDS, and for those undergoing cystoscopy, samples were collected before the procedure. Urine was collected by self-voiding when the patients experienced a full bladder sensation. Urinalysis was performed to confirm an infection-free status before storing the samples. A total of 50 mL of urine was placed on ice immediately and transferred to the laboratory. Samples were centrifuged at 1800 rpm for 10 min at 4 °C. The supernatant was aliquoted into 1.5 mL tubes and stored at −80 °C. Before analysis, frozen samples were centrifuged at 12,000 rpm for 15 min at 4 °C, and the supernatants were used for measurements.

The quantification of 8-OHdG, 8-isoprostane, and TAC in urine samples was performed using the respective ELISA kits (BioVision, Waltham, MA, USA; Enzo Life Science, Farmingdale, NY, USA; Abcam, Cambridge, UK). Inflammation-related cytokines were assayed using the Milliplex^®^ human cytokine/chemokine magnetic bead-based panel kit (Millipore, Darmstadt, Germany). Inflammatory analytes, including eotaxin, IL-2, IL-6, IL-8, IP-10, MCP-1, MIP-1β, RANTES, and TNF-α, were measured with the multiplex kit (catalog number HCYTMAG-60K-PX30). Additional neurogenic proteins analyzed included NGF, BDNF, and PGE2. Laboratory procedures followed previously reported methods [9,17,18,19].

### 4.4. Statistical Analysis

Urine biomarker levels were analyzed using descriptive statistics. The four groups included patients with NHIC, DO, or HSB, and controls. The median test was employed to identify significant differences in biomarker levels among the four groups due to the non-parametric nature of the data. For pairwise comparisons, the Dunn–Bonferroni post hoc test was used to control for multiple comparisons and to identify specific group differences. Receiver operating characteristic (ROC) curve analysis was performed to evaluate the diagnostic performance of individual urinary biomarkers and their combinations for distinguishing between the groups. The area under the ROC curve (AUROC) was calculated for each biomarker to quantify its diagnostic accuracy. Sensitivity, specificity, and the optimal cut-off values for the biomarkers were determined using the Youden index, which maximizes the sum of sensitivity and specificity. All statistical analyses were conducted using SPSS (Statistical Package for the Social Sciences) version 26.0 (IBM Corp., Armonk, NY, USA) and MedCalc Statistical Software version 19.6.4 (MedCalc Software Ltd., Ostend, Belgium). Statistical significance was set at *p* < 0.05 for all tests.

## 5. Conclusions

This study suggests that specific urinary biomarkers may hold significant diagnostic potential for differentiating between sensory bladder disorders in women. For example, TNF-α appears to be a reliable biomarker for identifying NHIC with high sensitivity and specificity, allowing for recommendations of cystoscopic hydrodistention under anesthesia for a faster diagnosis. While our findings present a robust indication that the combination of low urinary levels of 8-OHdG and TAC may serve as a useful diagnostic tool for HSB, further investigation is warranted.

Our findings highlight the possible role of non-invasive urinary biomarkers in enhancing the diagnosis and management of sensory bladder disorders, leading to more effective and timely treatment. Further research is needed to validate and expand these findings. 

## Figures and Tables

**Figure 1 ijms-25-09359-f001:**
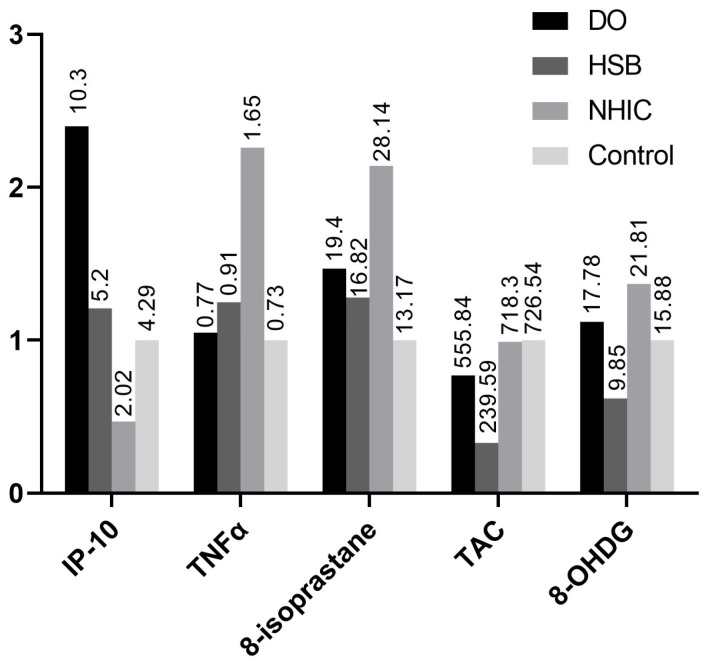
The ratio of each significant urine biomarker level in sensory bladder disorder groups to urodynamically normal control. The values at the top of each bar indicate the urine level for the individual biomarker. Abbreviations: DO—detrusor overactivity, NHIC—non-Hunner’s interstitial cystitis, and HSB—hypersensitive bladder.

**Figure 2 ijms-25-09359-f002:**
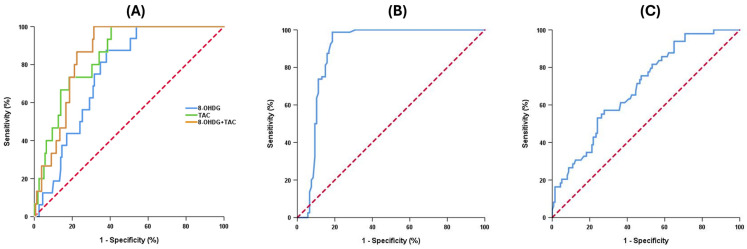
ROC curve analysis for urine cytokines. (**A**) HSB vs. total patients: ROC curves for 8-OHdG, TAC, and the combination of 8-OHdG + TAC. 8-OHdG shows an AUC of 0.754 with a cut-off value of ≤14.705, sensitivity of 87.5%, and specificity of 62.0%. TAC shows an AUC of 0.844 with a cut-off value of ≤528.720, sensitivity of 100%, and specificity of 59.5%. The combination of 8-OHdG and TAC shows an AUC of 0.853 with a cut-off value of ≤0.138, sensitivity of 100%, and specificity of 68.6%. The cut-off value for the combination was determined using a multiple regression model with the regression equation: y = 0.218 − 0.003 × 8-OHdG − 0.0000675 × TAC. Blue line: 8-OHdG; green line: TAC; orange line: 8-OHdG + TAC; red dash line: reference line. (**B**) NHIC vs. total patients: ROC curve for TNF-α with an AUC of 0.889, a cut-off value of ≥1.050, sensitivity of 98.8%, and specificity of 81.3%. Red dash line: reference line. (**C**) DO vs. total patients: ROC curve for IP-10 with an AUC of 0.695, a cut-off value of ≥6.310, sensitivity of 57.1%, and specificity of 72.3%. red dash line: reference line. Abbreviations: DO—detrusor overactivity, HSB—hypersensitive bladder, NHIC—non-Hunner type interstitial cystitis/bladder pain syndrome, TNF—tumor necrosis factor, TAC—total antioxidant capacity, and 8-OHdG—8-hydroxydeoxyguanosine.

**Figure 3 ijms-25-09359-f003:**
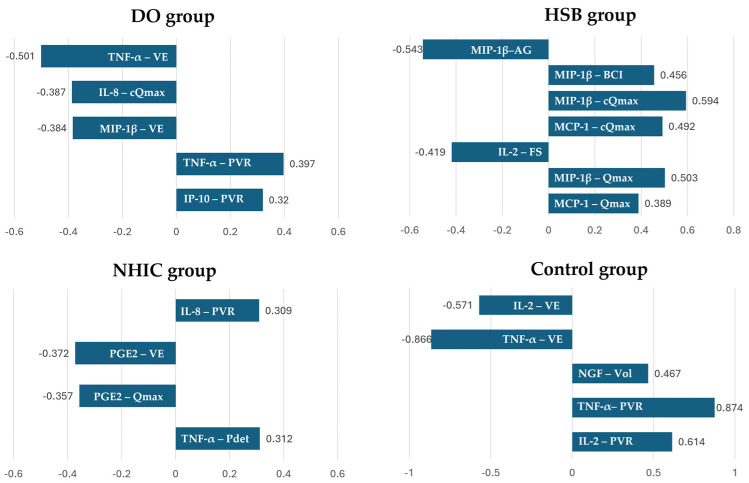
Correlations between biomarkers and VUDS parameters across four groups. Abbreviations: DO—detrusor overactivity, HSB—hypersensitive bladder, NHIC—non-Hunner type interstitial cystitis/bladder pain syndrome, Pdet—detrusor pressure, Qmax—maximum flow rate, PVR—post-void residual, FS—full sensation, Vol—bladder volume, BCI—bladder contractility index, cQmax—calculated as the product of Qmax and the square root of CBC, VE—voiding efficacy, AG—Abrams−Griffith number, calculated as Pdet at Qmax—2Qmax, IL—interleukin, TNF—tumor necrosis factor, NGF—nerve growth factor, and PGE2—prostaglandin E2.

**Table 1 ijms-25-09359-t001:** Urine biomarkers across sensory bladder disorder groups (DO, HSB, NHIC, and urodynamically normal).

Urine Biomarkers	(1) DO N = 51	(2) HSB N = 29	(3) NHIC N = 81	(4) Control N = 30	*p*-Value ^a^	Post Hoc ^b^
Age (years)	63.8 ± 11.1	61.7 ± 12.6	54.4 ± 12.7	58.9 ± 10.8	<0.001	12 vs. 3
Eotaxin	2.88 (1.64, 4.94)	2.57 (1.27, 5.5)	2.95 (2.64, 5.27)	3.84 (2.5, 7.3)	0.196	
IL-2	0.64 (0.59, 0.74)	0.65 (0.57, 0.77)	0.2 (0.18, 0.25)	0.89 (0.64, 0.98)	<0.001	124 vs. 3 12 vs. 4
IL-6	0.99 (0.81, 1.74)	0.78 (0.59, 1.32)	0.51 (0.28, 1.43)	0.87 (0.65, 1.23)	0.008	1 vs. 3
IL-8	9.72 (3.84, 32.35)	10.01 (2.09, 25.48)	9.2 (1.6, 26.15)	6.47 (1.43, 18.13)	0.701	
IP-10	10.3 (2.79, 42.13)	5.2 (2.21, 22.75)	2.02 (1.3, 4.96)	4.29 (1.98, 34.06)	0.001	1 vs. 3
MCP-1	218.75 (61.43, 348.33)	102.87 (28.65, 257.62)	157.75 (66.3, 303.34)	162.82 (65.42, 252)	0.715	
MIP-1β	1.91 (0.96, 3.29)	1.31 (0.93, 2.5)	0.28 (0.18, 1.31)	1.88 (1.15, 3.81)	<0.001	124 vs. 3
RANTES	3.92 (2.42, 8.15)	4.7 (2.63, 8.64)	1.63 (0.64, 4.4)	4.91 (2.7, 8.64)	0.001	124 vs. 3
TNF-α	0.77 (0.64, 1.04)	0.91 (0.7, 1.08)	1.65 (1.41, 1.78)	0.73 (0.63, 1)	<0.001	124 vs. 3
NGF	0.23 (0.18, 0.28)	0.19 (0.17, 0.2)	0.16 (0.14, 0.18)	0.25 (0.21, 0.3)	<0.001	124 vs. 3 2 vs. 4
BDNF	0.53 (0.42, 0.69)	0.5 (0.45, 0.57)	0.55 (0.48, 0.68)	0.51 (0.46, 0.7)	0.149	
PGE2	211.05 (139.68, 358.36)	n.s.	172.7 (115.43, 282.79)	145.48 (101.03, 204.7)	0.077	
8-isoprastane	19.4 (9.48, 42.12)	16.82 (12.39, 30.77)	28.14 (10.02, 58.6)	13.17 (7.45, 21.99)	0.083	
TAC	555.84 (256.47, 1391.78)	239.59 (148.58, 435.71)	718.3 (383.45, 1267.68)	726.54 (511.48, 1658.85)	0.001	2 vs. 34
8-OHdG	17.78 (7.76, 36.42)	9.85 (4.38, 13.49)	21.81 (11.69, 40.27)	15.88 (7.14, 30.06)	0.005	2 vs. 13

Values represent medians with interquartile ranges in parentheses; ^a^ statistical analysis among all subgroups; and ^b^ significant difference in post hoc analysis among subgroups. Abbreviations: DO—detrusor overactivity, HSB—hypersensitive bladder, NHIC—non-Hunner type interstitial cystitis/bladder pain syndrome, IL—interleukin, TNF—tumor necrosis factor, NGF—nerve growth factor, BDNF—brain-derived neurotrophic factor, PGE2—prostaglandin E2, TAC—total antioxidant capacity, and 8-OHdG—8-hydroxydeoxyguanosine.

**Table 2 ijms-25-09359-t002:** VUDS parameters across sensory bladder disorder groups (DO, HSB, NHIC, and urodynamically normal).

VUDS Parameters	(1) DO N = 51	(2) HSB N = 29	(3) NHIC N = 81	(4) Control N = 30	*p*-Value ^a^	Post Hoc ^b^
Pdet (cmH_2_O)	23.64 ± 17.04	16.32 ± 6.92	18.88 ± 11.95	15.52 ± 6.41	0.007	1 vs. 4
Qmax (mL/s)	12.95 ± 6.99	13.39 ± 5.77	11.61 ± 7.47	19.48 ± 8.14	<0.001	123 vs. 4
Volume (mL)	221.67 ± 125.24	264.68 ± 101.62	242.01 ± 137.79	429.39 ± 141.96	<0.001	123 vs. 4
PVR (mL)	19.58 ± 44.67	39.64 ± 69.25	52.63 ± 96.33	18.93 ± 75.54	0.049	
FSF (mL)	99.67 ± 45.82	179.93 ± 206.93	131.24 ± 52.32	165.9 ± 65.51	0.001	1 vs. 24 2 vs. 3
FS (mL)	158.9 ± 73.33	238.93 ± 52.92	198.4 ± 70.51	284.37 ± 95.96	<0.001	1 vs. 234 24 vs. 3
Compliance (mL/cmH_2_O)	56.62 ± 51.77	115.96 ± 61.89	63.98 ± 60.53	166.12 ± 105.29	<0.001	13 vs. 24
BCI	87.66 ± 33.47	86 ± 27.58	77.7 ± 37.75	116.65 ± 32.62	<0.001	123 vs. 4
CBC (mL)	244.79 ± 124.82	313.93 ± 75.07	294.91 ± 126.79	445.44 ± 117.8	<0.001	1 vs. 234 2 vs. 34 3 vs. 4
cQmax	0.86 ± 0.35	0.78 ± 0.3	0.68 ± 0.41	0.93 ± 0.35	0.009	14 vs. 3
VE	0.93 ± 0.14	0.87 ± 0.22	0.79 ± 0.32	0.95 ± 0.18	0.001	14 vs. 3
AG number	−1.97 ± 25.33	−10.7 ± 12.73	−4.64 ± 19.9	−25.48 ± 13.76	<0.001	123 vs. 4

^a^ Statistical analysis among all subgroups; and ^b^ significant difference in post hoc analysis among subgroups. Abbreviations: VUDS—videourodynamic study, DO—detrusor overactivity, HSB—hypersensitive bladder, NHIC—non-Hunner type interstitial cystitis/bladder pain syndrome, Pdet—detrusor pressure, Qmax—maximum flow rate, PVR—post-void residual urine amount, FSF—first sensation of filling, FS—full sensation, CBC—cystometric bladder capacity, BCI—bladder contractility index, cQmax—calculated as the product of Qmax and the square root of CBC, VE—voiding efficacy, AG number—Abrams–Griffith number, calculated as Pdet at Qmax—2Qmax.

## Data Availability

The data used in this study are available upon request from the corresponding author.

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
