# Peer review of "Using Urine Biomarkers to Differentiate Bladder Dysfunctions in Women with Sensory Bladder Disorders"

_ijms, 2024, doi:10.3390/ijms25179359_

Round 1

Reviewer 1 Report

Comments and Suggestions for Authors

The manuscript addresses an interesting and clinically relevant issue of differentiating between various types of sensory bladder disorders (SBD) such as non-Hunner’s interstitial cystitis (NHIC), detrusor overactivity (DO), hypersensitive bladder (HSB), and controls, using non-invasive urine biomarkers. The study's unique contribution lies in its focus on identifying specific biomarkers like TNF-α, IP-10, TAC, and 8-OHdG, which can aid in accurately diagnosing these conditions, potentially improving patient management by enabling tailored treatment strategies.

Areas for Improvement:

  1. Clarity and Depth of the Introduction:
    • The introduction effectively outlines the clinical significance and challenges associated with diagnosing SBD due to overlapping symptoms. However, it could be enhanced by providing a more detailed discussion on how current diagnostic methods fail to address these challenges and how the proposed biomarkers fill this gap. In addition, as you explore the utility of urinary biomarkers for diagnosing sensory bladder disorders, it is imperative to acknowledge their broader applications in detecting other urological conditions. A pertinent study (PMID: 38298766) investigates the relationship between the urinary microbiome and bladder cancer, revealing specific bacterial signatures associated with the disease. This study not only underscores the diagnostic potential of urinary biomarkers but also their predictive capabilities in identifying cancer risks. Such findings suggest that the urinary biomarkers we discuss could be part of a larger panel that assists in comprehensive urological assessments, bridging the gap between diagnosing sensory disorders and identifying oncological risks. Please cite and include the suggested article. 
  2. Methodology:
    • The study design and data analysis are well-detailed, yet the manuscript would benefit from a clearer explanation of the selection criteria for the participants and more rigorous validation of the biomarkers across different patient populations.
  3. Results Presentation:
    • The results are promising but presented in a manner that could be simplified for better understanding. More visual aids (charts or graphs) summarizing the findings could make the data more accessible to readers.
  4. Discussion and Interpretation:
    • While the discussion highlights the potential utility of the identified biomarkers, it lacks a critical analysis comparing these findings with existing literature. Including such a comparison could strengthen the argument for the biomarkers' diagnostic power. In exploring diagnostic biomarkers for sensory bladder disorders, it is pertinent to consider novel therapeutic interventions that may interact with or complement these biomarkers in clinical practice. The following study PMID: 34680774 reviews the potential of intravesical platelet-rich plasma (PRP) injections in treating bladder pain syndrome/interstitial cystitis (BPS/IC), conditions closely related to the disorders under investigation in our study. This treatment shows promise in enhancing urothelial regeneration and reducing chronic inflammation, key factors that are also potentially modifiable by the urinary biomarkers we have identified. Such therapeutic advancements underscore the need for integrated diagnostic and treatment strategies, enriching the clinical relevance of our findings in urinary biomarkers.
  5. Statistical Analysis:
    • The statistical methods used are appropriate.
  6. Clinical Relevance:
    • The clinical implications of the findings are mentioned, but the manuscript would benefit from a more detailed discussion on how these biomarkers can be integrated into current clinical workflows.
  7. Limitations:
    • The study acknowledges several limitations, such as the lack of comorbidity control and the exclusion of men, which could affect the generalizability of the findings. Addressing how these limitations could be overcome in future studies would provide a more balanced view.

Author Response

Reviewer 1

Comments 1: Clarity and Depth of the Introduction:

The introduction effectively outlines the clinical significance and challenges associated with diagnosing SBD due to overlapping symptoms. However, it could be enhanced by providing a more detailed discussion on how current diagnostic methods fail to address these challenges and how the proposed biomarkers fill this gap. In addition, as you explore the utility of urinary biomarkers for diagnosing sensory bladder disorders, it is imperative to acknowledge their broader applications in detecting other urological conditions. A pertinent study (PMID: 38298766) investigates the relationship between the urinary microbiome and bladder cancer, revealing specific bacterial signatures associated with the disease. This study not only underscores the diagnostic potential of urinary biomarkers but also their predictive capabilities in identifying cancer risks. Such findings suggest that the urinary biomarkers we discuss could be part of a larger panel that assists in comprehensive urological assessments, bridging the gap between diagnosing sensory disorders and identifying oncological risks. Please cite and include the suggested article.

Response 1: Thank you very much. We have enhanced the introduction in the revised manuscript by detailing the limitations of current diagnostic methods for sensory bladder disorders (SBD). Specifically, we discussed how these methods focus largely on structural assessments and often necessitate invasive procedures, which fail to capture the biochemical differences necessary for accurate differentiation (Line: 44-57).

We incorporated a discussion on the potential role of urinary biomarkers not only in diagnosing SBD but also in broader urological applications. We cited the study with PMID: 38298766, which demonstrates the relevance of urinary microbiomes in bladder cancer, to reinforce the broader diagnostic and predictive capabilities of urinary biomarkers. This inclusion highlights the potential for these biomarkers to assist in comprehensive urological assessments (Line: 67-75).

Comments 2: Methodology:

The study design and data analysis are well-detailed, yet the manuscript would benefit from a clearer explanation of the selection criteria for the participants and more rigorous validation of the biomarkers across different patient populations.

 Response 2:

Thank you for highlighting the need for clarity regarding participant selection criteria and biomarker validation. We have added specific details about our inclusion and exclusion criteria to enhance understanding of how the study cohort was defined. This ensures focus on relevant sensory bladder disorders without confounding factors. (Line: 340-345)

Additionally, while our current study involved a single-center cohort, we acknowledge the importance of validating these biomarkers across more diverse populations in the discussion section in the revised manuscript (Line: 331-333).

The causes of LUTS differ between males and females, so this study focuses solely on females. Future studies will aim to validate these findings across a multiregional cohort that includes both male and female participants (Line 326-328). This approach will facilitate a deeper understanding of the biomarkers' applicability across diverse populations and urological settings, confirming their role in routine clinical practice.

Comments 3: Results Presentation:

The results are promising but presented in a manner that could be simplified for better understanding. More visual aids (charts or graphs) summarizing the findings could make the data more accessible to readers.

Response 3: Thank you for your valuable feedback on the presentation of our results. We have simplified Table 4 into a more accessible format by converting it into Figure 3 in the revised manuscript (Line 159-167). This graphical representation will help present our findings more clearly and enable readers to grasp the data more easily through visual aids.

Comments 4: Discussion and Interpretation:

While the discussion highlights the potential utility of the identified biomarkers, it lacks a critical analysis comparing these findings with existing literature. Including such a comparison could strengthen the argument for the biomarkers' diagnostic power. In exploring diagnostic biomarkers for sensory bladder disorders, it is pertinent to consider novel therapeutic interventions that may interact with or complement these biomarkers in clinical practice. The following study PMID: 34680774 reviews the potential of intravesical platelet-rich plasma (PRP) injections in treating bladder pain syndrome/interstitial cystitis (BPS/IC), conditions closely related to the disorders under investigation in our study. This treatment shows promise in enhancing urothelial regeneration and reducing chronic inflammation, key factors that are also potentially modifiable by the urinary biomarkers we have identified. Such therapeutic advancements underscore the need for integrated diagnostic and treatment strategies, enriching the clinical relevance of our findings in urinary biomarkers.

Response 4: Thank you for your valuable feedback regarding the discussion and interpretation of our findings. In revised manuscript, we have incorporated a critical analysis comparing our results with existing literature (Line 238-243) and discussed the potential therapeutic interventions that interact with or complement the biomarkers we identified (introduced the relevance of treatments like PRP injections with the reference) (Line 244-252).

Comments 5: Statistical Analysis: The statistical methods used are appropriate.

Response 5: Thank you very much.

Comments 6: Clinical Relevance:

The clinical implications of the findings are mentioned, but the manuscript would benefit from a more detailed discussion on how these biomarkers can be integrated into current clinical workflows.

Response 6: Thank you very much. Our findings reveal distinct inflammatory profiles and oxidative stress biomarkers among women with different sensory bladder disorder and addresses this need by seeking more accurate predictive biomarkers that can be applied in clinical settings to distinguish between bladder sensory disorders presenting with similar symptoms. By focusing on this specific group, we aimed to identify urinary biomarkers that could differentiate between these sensory bladder disorders and provide greater clinical utility. The findings of this study emphasize the diagnostic potential of specific urinary biomarkers, such as TNF-α, IL-2, 8-OHdG, and TAC, in managing sensory bladder disorders. Integrating these biomarkers into current clinical workflows could significantly enhance the accuracy of diagnosis and the effectiveness of treatment strategies in the future. We have added more discussion in the revised manuscript (Line: 223-229).

Comments 7: Limitations

The study acknowledges several limitations, such as the lack of comorbidity control and the exclusion of men, which could affect the generalizability of the findings. Addressing how these limitations could be overcome in future studies would provide a more balanced view.

Response 7: Thank you very much. The limitations of this study are disclosed in the last part of the discussion (Line 324-335).

Reviewer 2 Report

Comments and Suggestions for Authors

Dear authors,

Thank you for your valuable manuscript. However, there are some issues to be fixed before a possible publication.

- you are trying to introduce the value of some biomarkers, talking about female LUTD. However, you include patients with HSB and NHIC (definite clinical conditions), while you also study those with DO (urodynamic finding). It would be more scientifically accurate, if you use only clinical situations, as your study title says ("sensory bladder disorders").

- it is not clear, how you recruited patients. Did you use bladder diaries? If not, how did you evaluate the functional disorders before VUDS? If it only by medical history, it could be regarded as biased.

- your conclusions should be less direct.

Author Response

Reviewer 2

Comment 1:

You are trying to introduce the value of some biomarkers, talking about female LUTD. However, you include patients with HSB and NHIC (definite clinical conditions), while you also study those with DO (urodynamic finding). It would be more scientifically accurate, if you use only clinical situations, as your study title says ("sensory bladder disorders").

Response 1:

Thank you very much for your insightful feedback regarding our study design. We acknowledge that including patients with detrusor overactivity (DO), which is a urodynamic finding rather than a strictly defined clinical condition like hypersensitive bladder (HSB) and non-Hunner type interstitial cystitis (NHIC), may complicate the interpretation of our results.

However, our study addresses this issue by recognizing that at the time of initial consultation, symptoms for DO, HSB, and NHIC all manifest as sensory bladder symptoms, including storage LUTS and bladder-related discomfort. These sensory experiences are inherently subjective and can be difficult to differentiate clearly at the outset. Therefore, by utilizing objective urine biomarkers, we aim to clarify the role of these biomarkers in sensory bladder disorders—specifically, whether they can help us discern between different conditions earlier on and facilitate more precise (personalized) treatment options.

Although the clinical management of DO and HSB may not differ significantly, our study can provide clinicians with additional information that will also aid future papers in further elucidating the underlying mechanisms of each disorder.

Thank you once again for your valuable insights.

Comment 2

It is not clear, how you recruited patients. Did you use bladder diaries? If not, how did you evaluate the functional disorders before VUDS? If it only by medical history, it could be regarded as biased.

Response 2: Thank you for the comment. All patients were requested to keep a three-day voiding diary and record the daytime and night time voiding frequency, voided volume, episodes of urgency and urgency incontinence (UUI). The maximal voided volume was recorded as the functional bladder capacity (FBC). When patients had an average daily frequency of more than 8 times and FBC less than 350ml, bladder hypersensitivity was considered. When there were episodes of urgency or UUI, overactive bladder was diagnosed. (Line: 346~353)

All patients underwent videourodynamic study (VUDS) for the diagnosis of bladder dysfunction including detrusor overactivity (DO), detrusor underactivity (DU), or bladder outlet obstruction (BOO) based on the pressure flow study and voiding cystourethrography features. (Line 356~365)

The diagnostic criteria were in accordance with the recommendations of ICS/SUFU guidelines. In addition, a potassium chloride (KCl) test was routinely performed after VUDS and patients with a positive test were advised to undergo cystoscopic hydrodistention under intravenous general anesthesia for the diagnosis of interstitial cystitis/bladder pain syndrome (IC/BPS). (Line 369-371)

We have added them in the revised manuscript.

Comment 3

Your conclusions should be less direct.

Response 3:

Thank you very much. We have revised the conclusion section to adopt a more nuanced tone. We have emphasized the suggestive nature of our findings concerning the diagnostic potential of urinary biomarkers without making overly definitive claims. This adjustment aims to reflect a more exploratory approach while still highlighting the significance of our research. (Line 435-444)

Round 2

Reviewer 1 Report

Comments and Suggestions for Authors

The manuscript has been significantly improved from the previous version, and the revisions have clearly strengthened the work. The authors have successfully addressed the previous comments, enhancing the clarity, focus, critical analysis, and integration of the literature. With these revisions, the article now represents a valuable contribution to the field of bladder cancer research and is worthy of publication.

Reviewer 2 Report

Comments and Suggestions for Authors

Accepted in the revised form.